# Case Report: Long-Term Survival of a Patient with Cerebral Metastasized Ovarian Carcinoma Treated with a Personalized Peptide Vaccine and Anti-PD-1 Therapy

**DOI:** 10.3390/vaccines12040397

**Published:** 2024-04-09

**Authors:** Henning Zelba, Christina Kyzirakos, Simone Kayser, Borong Shao, Annekathrin Reinhardt, Natalia Pieper, Armin Rabsteyn, Dennis Döcker, Sorin Armeanu-Ebinger, Matthias Kloor, Dirk Hadaschik, Martin Schulze, Florian Battke, Alexander Golf, Saskia Biskup

**Affiliations:** 1Zentrum für Humangenetik Tübingen, 72076 Tübingen, Germany; 2Institut für Medizinische Genetik und Angewandte Genomik, Universitätsklinikum Tübingen, 72076 Tübingen, Germany; 3Institut für Pathologie, Heidelberg University Hospital, 69120 Heidelberg, Germany; 4CeCaVa GmbH & Co. KG, 72076 Tübingen, Germany; 5CeGaT GmbH, 72076 Tübingen, Germany; 6MVZ Zentrum für ambulante Onkologie GmbH, 72076 Tübingen, Germany

**Keywords:** personalized peptide vaccination, ovarian cancer, Beta-2 microglobulin, *MKKS*

## Abstract

Ovarian cancer is one of the most common cancers among women and the most lethal malignancy of all gynecological cancers. Surgery is promising in the early stages; however, most patients are first diagnosed in the advanced stages, where treatment options are limited. Here, we present a 49-year-old patient who was first diagnosed with stage III ovarian cancer. After the tumor progressed several times under guideline therapies with no more treatment options available at that time, the patient received a fully individualized neoantigen-derived peptide vaccine in the setting of an individual healing attempt. The tumor was analyzed for somatic mutations via whole exome sequencing and potential neoepitopes were vaccinated over a period of 50 months. During vaccination, the patient additionally received anti-PD-1 therapy to prevent further disease progression. Vaccine-induced T-cell responses were detected using intracellular cytokine staining. After eleven days of in vitro expansion, four T-cell activation markers (namely IFN-ɣ, TNF-α, IL-2, and CD154) were measured. The proliferation capacity of neoantigen-specific T-cells was determined using a CFSE proliferation assay. Immune monitoring revealed a very strong CD4+ T-cell response against one of the vaccinated peptides. The vaccine-induced T-cells simultaneously expressed CD154, TNF, IL-2, and IFN-ɣ and showed a strong proliferation capacity upon neoantigen stimulation. Next-generation sequencing, as well as immunohistochemical analysis, revealed a loss of Beta-2 microglobulin (B2M), which is essential for MHC class I presentation. The results presented here implicate that the application of neoantigen-derived peptide vaccines might be considered for those cancer stages, where promising therapeutic options are lacking. Furthermore, we provide more data that endorse the intensive investigation of B2M loss as a tumor escape mechanism in clinical trials using anti-cancer vaccines together with immune-checkpoint inhibitors.

## 1. Introduction

Ovarian cancer (OC) originates from cells of the ovary, peritoneum, or the fallopian tube. OC is one of the most common cancers among women and the most lethal malignancy of all gynecological cancers [1,2]. Precisely, type I epithelial OCs are suggested to be relatively indolent and genetically stable tumors that typically arise from recognizable precursor lesions, such as endometriosis or borderline tumors with low malignant potential. In contrast, type II epithelial OCs are proposed to be biologically aggressive tumors from their outset, with a propensity for metastasis from small-volume primary lesions. High-grade serous, which is the most common type of epithelial OC—accounting for approximately 75% of all epithelial ovarian cancers—develop according to the type II pathway and present p53 and BRCA mutations [3]. The current standard of care of early stage OC is primary debulking surgery (PDS). If a complete resection of the tumor is not possible, neoadjuvant chemotherapy followed by interval debulking surgery (IDS) or palliative treatment is recommended. For advanced-stage OC, a debulking procedure in combination with chemotherapy is the standard treatment [4].

After chemotherapy, there are several options of maintenance therapy, like anti-angiogenic agents (e.g., bevacizumab; [5]) or PARP (poly ADP ribose polymerase) inhibitors (e.g., Olaparib; [6]).

While the five year survival rate for stage I is about 90%, it is below 20% for stage IV [7,8]. Unfortunately, most patients are first diagnosed in the advanced stages (stage 2b or higher), due to the lack of initial symptoms and extensively applied screening methods [8]. For the late stages, median overall survival is reported to be about 34 months [9]. As a result, most patients diagnosed with OC face a relatively poor prognosis, indicating that there is an urgent need for advanced therapeutic options.

Immunotherapy is considered as the fourth pillar of cancer treatment. However, many cancer patients still do not benefit from approved immunotherapeutic approaches like immune-checkpoint inhibitors. For OC, although promising results were obtained in preclinical and early phase clinical studies (e.g., combinations of PARP inhibitors with immunotherapies, such as anti-CTLA-4 and PD-1/PD-L1), the use of immunotherapeutic treatments has still not been implemented in clinical routine, mostly due to insufficient evidence of clinical effectiveness. Neither single [10] nor combined immune-checkpoint inhibitors [11], nor off-the-shelf vaccines [12], have shown significant clinical efficacy so far. Thus, personalized treatment regimens and a combination of several therapies are currently an intensively investigated research area.

We present a patient with cerebral metastasized ovarian carcinoma. The incidence of brain metastases in OC is rare, reported to range between 0.5% and 6.1%. There are no established guidelines for brain metastases, hence prognosis is poor with a median overall survival of about 10 months [13]. In recurrent brain metastases, the prognosis is even worse.

Here, we report the treatment with an individualized neoantigen-derived peptide vaccination, combined with anti-PD-1 therapy in the setting of an individual healing attempt.

## 2. Case Description

We report the case of a 49-year-old female patient who was diagnosed with metastasized ovarian cancer in November 2010 (pT3b, nN1, M0; serous adenocarcinoma grading G3) (Figure 1). After initial diagnosis, she underwent PDS (tissue 1; primary tumor) followed by standard chemotherapy. After a first (2011), second (February 2014; tissue 2; spleen metastasis), and third progression (September 2014; tissue 3; lymph node metastasis), each, subsequently, treated according to guidelines, the patient was proposed for experimental personalized peptide vaccination in a single patient compassionate use healing attempt. Briefly, the lymph node metastasis (tissue 3) was analyzed for somatic mis-sense variants using exome sequencing and was used for an in silico epitope prediction of mutated peptides possibly binding to the specific HLA molecules of the patient. A total of 57 tumor-specific mis-sense variants were identified and led to the final selection and synthesis of five peptides (see Table 1). Peptides were synthesized and formulated into a vaccine cocktail. For each vaccination, 0.5 mL multipeptide solution was injected intracutaneously into the lower abdomen, followed by subcutaneous injection of 250 µg sargramostim (GM-CSF) at the vaccination site. The patient was first vaccinated in December 2014.

Three months after the first vaccination, three brain metastases were detected using PET/CT scans (March 2015). They were treated with cyberknife radiation (3 cycles, 1 × 18 Gy, 1 × 20 Gy). The vaccination schedule was continued. Three months later (June 2015), a PET/CT of the brain showed only one residual lesion with regredient size (partial response; PR).

Ten days later, about six months after vaccine start, anti-PD-1 therapy (Pembrolizumab) was combined in off-label use with the vaccination (4 doses; every 3 or 4 weeks). Due to a pronounced erythema observed at the vaccination site after the 10th vaccination, the following two injections were followed by the superficial application of imiquimod crème and a lower dose of sargramostim (83 µg) (Appendix A). No systemic side effects were observed at that time.

In August 2015, the patient presented with neurological symptoms (cognitive impairment, right hemiparesis). Pembrolizumab treatment was stopped. A cranial MRT revealed the progression of the residual lesion and the occurrence of one new, smaller metastasis. The larger lesion was surgically resected (R0; tissue 4); the smaller lesion was treated with radiotherapy (both in September 2015). A subsequent MRI in October 2015 was interpreted as pseudoprogression. Brain metastasis tissue was used for an additional sequencing analysis and resulted in the addition of another peptide to the cocktail.

The last vaccination took place in February 2019. From October 2015 to December 2019, the patient remained in a stable disease state of the scars of the brain metastases and remission in the rest of the body, as documented using repeated imaging including PET-CTs and brain MRI scans.

In April 2020, the patient was diagnosed with a radiation-induced synovial sarcoma in the right hip, due to extensive radiation therapy. The patient deceased in December 2020 due to disease progression of the sarcoma.

## 3. Materials and Methods

### 3.1. Vaccination

Vaccine design, formulation, and administration was performed as previously described [14]. Briefly, selected peptides were synthesized and formulated into vaccines. For each vaccination, 0.5 mL multipeptide solution (0.8 mg/mL per peptide) was injected intracutaneously into the lower abdomen, followed by subcutaneous injection of 250 µg sargramostim at the vaccination site. The patient was first vaccinated in December 2014, starting with a one month priming phase (vaccinations on days 1, 3, 8, 15, and 29), followed by a boosting phase of monthly injections. One additional longer peptide was vaccinated from 03/2016 to 02/2018 (18 vaccinations). To assess skin responses to the vaccination regime, the skin was monitored for erythema immediately after the injection and 1 h later. After the 10th injection, Imiquimod (topical cream, Aldara^®^ 5%) was used two times as a vaccine adjuvant and a lower dose of sargramostim (83 µg/injection) was used afterwards. Formulation and administration were performed at the Zentrum für Humangenetik Tübingen, Germany. In total, the patient received 47 vaccine doses in 4 years and 2 months.

### 3.2. Detection of Antigen-Specific T-Cell Responses

Peptide-specific T-cell responses were assessed after an in vitro pre-amplification protocol using intracellular cytokine staining (ICS). Briefly, peripheral blood mononuclear cells (PBMCs) were isolated using Ficoll Hypaque (Biochrom, Berlin, Germany) density gradient centrifugation from heparinized blood and were cryopreserved in 90% inactivated FCS (Thermo Scientific, Dreieich, Germany) + 10% DMSO (WAK Chemie, Steinbach, Germany) in liquid nitrogen for later use.

For the 11 day in vitro pre-sensitization, PBMCs were thawed at 37 °C, washed, and incubated in medium with 3 µg/mL Pulmozyme (Roche, Austria) at a concentration of 1 × 10^7^ cells/mL (day −1) and preincubated in TexMACS™ serum-free medium (Miltenyi Biotech, Bergisch Gladbach, Germany). On the next day (day 0), cells were washed with medium, counted using trypane blue staining, and seeded in a 48-well plate at 5 × 10^6^ cells/mL medium (250 µL/well). Peptides were added at a final concentration of 1 µg/mL in duplicate wells. On the next day (day 1) 250 µL medium/well was added (day 1), supplemented with IL-7 (final concentration 10 ng/mL, CellGenix, Freiburg, Germany) and IL-2 (final concentration 10 U/mL, Novartis, Nürnberg, Germany). Half of the medium was replaced with fresh medium incl. cytokines on days 4, 6, and 8. On day 9, 500 µL medium was added. On day 10, 750 µL medium was replaced. On day 11, medium was removed and cells were re-stimulated in 250 µL cell suspension. Each duplicate well was restimulated with 1 µg/mL of the respective peptide or aqua ad iniectabilia diluted DMSO at an equal volume (background control). As positive controls, unstimulated wells were stimulated with a mixture of viral overlapping peptide pools (EBV consensus, CMV pp65, and AdV5 Hexon; Miltenyi Biotec, Germany; 1 μg/mL each) and Staphylococcus enterotoxin B (SEB) (Sigma-Aldrich, Steinheim, Germany; 10 µL/mL). After 2 h of incubation, Brefeldin A (Sigma-Aldrich, Steinheim, Germany; 10 µg/mL) was added to the culture and incubated for an additional 4 h. Cells were collected and extracellular antibody staining was performed by adding anti-CD56, anti-CD3, anti-CD4, and anti-CD8 antibodies. Thereafter, cells were stained with AF350 to exclude dead cells. After fixation and permeabilization, cells were intracellularly stained with anti-IL-2, anti-IFN-γ, and anti-CD154 antibodies. Cells were measured on a LSRII cytometer (BD Bioscience, Franklin Lakes, NJ, USA). The resulting data were analyzed using FlowJo software version 7.6.5 (FlowJo LLC, Ashland, OR, USA).

For the proliferation assay, PBMCs were thawed at day −1, as described above. Cells were washed and labeled with 1.6 μM CFSE (Invitrogen, Karlsruhe, Germany) and incubated at 37 °C and 5% CO_2_. After 9 min, the reaction was stopped by adding FCS for 2 min. Subsequently, the cells were washed in RPMI medium and 7.5 × 10^5^ cells/well were seeded in 48-well plates in TexMACS™ medium. Peptides were added at a final concentration of 1 µg/mL. In total, 3.75 × 10^5^ cells each were stimulated with SEB and viral overlapping peptide pools, as described above. For blocking experiments, anti-HLA-DR, DB, DQ antibody (BD Bioscience, Heidelberg, Germany) at a final concentration of 10 μg/mL was added at day 0 and for re-stimulation. No cytokines were added to the assay. After 6 days, cells were re-stimulated followed by intra- and extracellular antibody staining and analyzed as described above, with the exception that no anti-IL-2 staining was performed.

### 3.3. Immunohistochemistry

Tumor tissue sections (tissue 1–4) were stained using the Vectastain ABC detection system (Vector Laboratories, Newark, CA, USA), according to the manufacturer’s instructions. Briefly, 2 µm sections were deparaffinized using xylene and decreasing concentrations of ethanol. Subsequently, antigen retrieval was performed by heating the slides in a microwave oven (3 times for 5 min at 560 W) at pH 6.0. Tissue sections were incubated with 6.7 µg/mL of the mouse monoclonal antibody L368 (specific for B2M) and 5 µg of LGII-612.14 (specific for HLA-DR, -DP, and -DQ). After washing, the sections were incubated with biotinylated anti-mouse IgG as a secondary antibody (Vector Laboratories) for 1 h at room temperature and staining was visualized using diaminobenzidine (Dako Cytomation, Hovedstaden, Denmark) as the substrate.

## 4. Results

### 4.1. Immune Monitoring of Vaccine-Induced T-Cell Responses

Peptide-specific T-cell responses were assessed in a PBMC sample collected in November 2015 (16th vaccination) using an in vitro pre-amplification protocol, followed by intracellular cytokine staining (ICS). Immune monitoring revealed a strong CD4+ T-cell response (30.4% within all CD4+ T-cells) against peptide KQLHNGFGSY (*MKKS-*p.G52S). The cells expressed/produced mainly CD154, TNF, and IL-2 and, to a lesser extent, IFN-ɣ (Figure 2A,B). Additionally, a proliferation assay showed a high proliferative capacity and a high antigen specificity of the T-cells. After 6 days of in vitro cultivation, nearly 20% of all CD4+ T-cells had undergone cell division after stimulation with the mutated peptide KQLHNGFGSY; *MKKS-*p.G52S, while the percentage of self-dividing cells after stimulation with the respective wildtype peptide was similar to the unstimulated control. Furthermore, the addition of an anti-HLA-DQ/DP/DR antibody blocked the antigen recognition (Figure 2C).

Immune monitoring was conducted at later timepoints to monitor vaccine-specific T-cell responses. A robust peptide-specific CD4+ T-cell response against the peptide KQLHNGFGSY (*MKKS*-p.G52S) was confirmed in a PBMC sample from February 2017. Interestingly, in a PBMC sample from June 2018, in addition to the CD4+ T-cell response, a CD8+ T-cell response against peptide KQLHNGFGSY (*MKKS*-p.G52S) was detected. Furthermore, a CD4+ T-cell response to a second vaccinated peptide (PYIRTKLIY; *GAN*-p.N60I) became detectable.

### 4.2. HLA Expression

Exome sequencing data of the brain metastasis (August 2015; tissue 4) revealed a homozygous loss of the first exon of B2M (Beta-2 microglobulin), which is essential for HLA class I stabilization, and a haplo loss of the HLA I locus (loss of HLA-A*02:01, -B*51:01 and C*05:01). Immunohistochemistry confirmed the loss of B2M in the brain metastasis, but also in the retrospectively analyzed spleen metastasis (tissue 2) and, to a lesser extent, in the lymph node metastasis (tissue 3) from 2014. The primary tumor was not affected (tissue 1; Figure 3A). Because of the detection of vaccine-induced neoantigen-specific CD4+ T-cells, we also analyzed HLA II expression. Immunohistochemistry revealed expression of HLA II in the primary tumor (tissue 1) and both metastases taken before vaccination but a weak/partial expression in the brain metastasis (tissue 4) (Figure 3B).

## 5. Discussion

Immunotherapy has evolved as the fourth pillar of cancer treatment in recent years. However, many studies in recent years showed that only subgroups of patients are benefitting from immunotherapeutic approaches. Among many other strategies, personalized vaccination against tumor-specific (neo)antigens is currently an intensively investigated research area [15,16,17]. Here, we present a late-stage OC patient receiving individualized neoantigen-derived peptide vaccination in the setting of an individual treatment attempt. Although OC generally has a lower TMB [18], we were still able to identify potential neoantigens suitable for vaccination. Immune monitoring revealed a strong polyfunctional (cytokine production and proliferation) vaccine-induced CD4+ T-cell response against a peptide derived from the *MKKS*-p.G52S variant. Interestingly, it was shown in OC patients, that CD4+ tumor-infiltrating T-cells can recognize autologous tumor cells in an MHC class II-restricted fashion [19]. Using animal models, Kruse et al. showed that CD4+ T-cells are able to independently eradicate established tumors as efficiently as CD8+ T-cells [20].

After the patient still progressed under personalized vaccine treatment and anti-PD-1 therapy (Pembrolizumab), we hypothesized that the recurrent tumor could underlie some kind of immune escape. Indeed, next-generation sequencing revealed a loss of Beta-2 microglobulin (B2M) and confirmed that previously detected somatic mutations were still present in the brain metastasis. The B2M loss was confirmed using immunohistochemistry. Furthermore, HLA class I molecules are instable without B2M and, consequently, peptide presentation and recognition by cytotoxic T-cells is dependent on B2M expression. *B2M* gene mutations are enriched in uterine, breast, colorectal, and gastric cancer and accumulate during cancer progression and targeted treatment [21]. Accumulating evidence has shown that alterations of *B2M* gene and B2M proteins contribute to poor reaction to cancer immunotherapies by dampening antigen presentation [22]. Hence, B2M is a potential biomarker for cancer immunotherapy [23]. In consensus with our findings, Zaretsky et al. identified a truncating mutation in the *B2M* gene, which led to the loss of surface expression of HLA I in a metastatic melanoma patient who had had an initial objective tumor regression in response to anti–PD-1 therapy, followed by disease progression [24]. Hence, we provide further data that endorse the intensive investigation of B2M in clinical trials applying anti-cancer vaccines and/or immune-checkpoint inhibitors.

During ongoing treatment, the vaccine-induced immune response remained robust and durable. Although the patient received multiple therapies and the clinical benefit cannot ultimately be attributed to neoantigen-derived peptide vaccination, strikingly, the patient remained progression free for more than four years. Later, the patient was unfortunately diagnosed with a radiation-induced synovial sarcoma, due to extensive radiation therapy. The patient deceased in December 2020 due to disease progression of the sarcoma, resulting in an overall survival time of 120 months.

We noticed no severe side effects due to vaccination. Minor temporal local skin reactions at vaccination sites such as redness, swelling, and erythema resolved without interventions. This is in line with clinical trials using peptide-based vaccines combined with sargramostim [25,26].

Taken together, we provide more evidence that it is technically feasible to produce a fully individualized neoantigen-derived peptide vaccine. Its application was well tolerated and immunogenic. We previously observed similar results in a urothelial carcinoma patient [27], a metastatic castration-sensitive prostate cancer patient [28], as well as four breast cancer patients [14].

There are several limitations that should be addressed in future studies. These include the ideal vaccine and adjuvants scheduling, the optimal timing of a combined immune-checkpoint inhibition, as well as the role of B2M loss in disease progression. If more promising data can be obtained, peptide vaccines might have the potential to be used as an interventional treatment in newly diagnosed OC patients.

## Figures and Tables

**Figure 1 vaccines-12-00397-f001:**
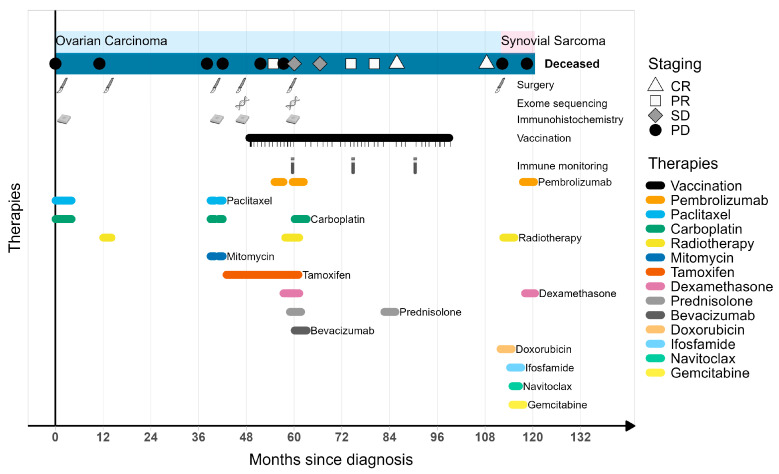
Clinical course of the patient since first diagnosis (November 2010). CR = complete response; PR = partial response; SD = stable disease; PD = progressive disease.

**Figure 2 vaccines-12-00397-f002:**
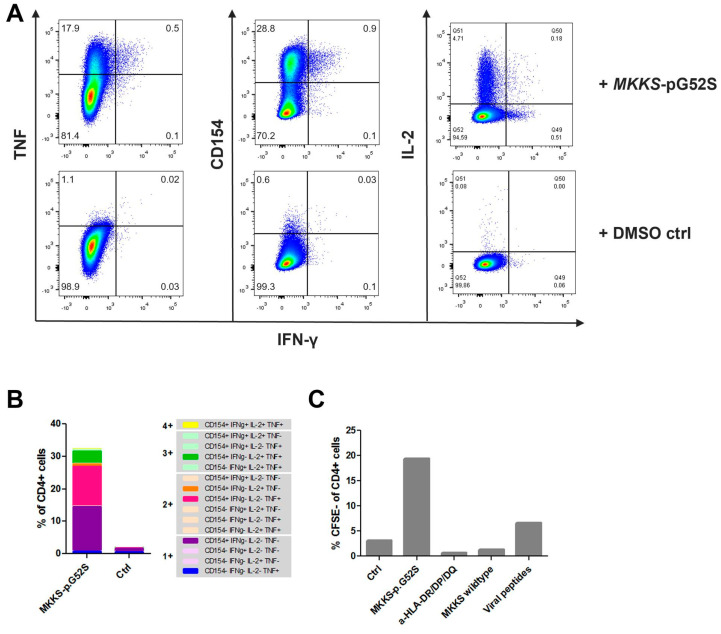
Immune monitoring. (**A**,**B**) CD4+ specific T-cell response towards KQLHNGFGSY (*MKKS*-p.G52S) after pre-stimulation in vitro. (**A**) TNF-, CD154-, IL-2-, and IFN-γ-positive cells after stimulation with *MKKS*-p.G52S peptide (upper row) versus DMSO control (lower row). (**B**) Boolean gate analysis of polyfunctionality of responding CD4+ T-cells. Major populations are highlighted (**C**) Proliferation assay. Cells are stimulated with DMSO control, *MKKS*-p.G52S peptide (KQLHNGFGSY), *MKKS*-p.G52S peptide + anti-HLA-DR/DP/DQ mAb, *MKKS*-wildtype peptide (KQLHNGFGGY), or a viral peptide mix.

**Figure 3 vaccines-12-00397-f003:**
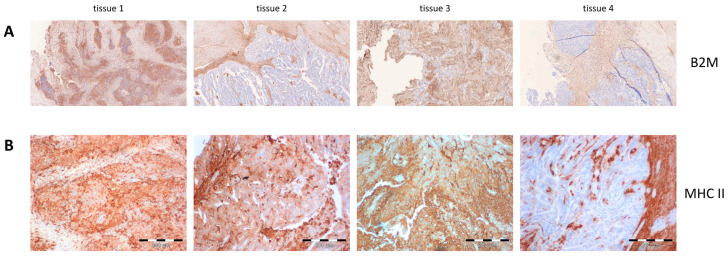
Immunohistochemistry. Expression analysis of Beta-2 microglobulin (**A**) and MHC II (**B**). Tissue 1: primary tumor (December 2010); tissue 2: spleen metastasis (February 2014); tissue 3: lymph node metastasis (September 2014); tissue 4: brain metastasis (September 2015).

**Table 1 vaccines-12-00397-t001:** Peptides selected for vaccination.

Neoantigenic Peptide	Gene and Variant	Predicted HLA Restriction	Variant Allele Frequency in the Lymph Node Metastasis (2014; Tissue 3)	Variant Allele Frequency in the Brain Metastasis (2015; Tissue 4)
PYRTKIY	*GAN*, NM_022041.3: c.179A>T; p.N60I	C*05:01	0.94	0.94
KQLHNGFGSY	*MKKS*, NM_018848.3: c.154G>A; p.G52S	A*32:01, C*14:02	0.76	1.00
SMSAETMEL	*BTN3A2*, NM_001197247.2: c.170C>G; p.T57S	A*02:01, A*32:01,	0.50	0.81
AETANLEEQL	*ARL6IP1*, NM_015161.1: c.53G>A; p.S18N	C*14:02	0.46	0.93
DPSTLGSL	*SYT6*, NM_205848.3: c.125T>C; p.F42S	B*44:02	0.32	0.30
EHAKAVVSFRHIQRRAA	*CCHCR1*, NM_001105564:c.A2178C:p.L726F (somatic) and c.2184G>T:p.Q728H (single nucleotide polymorphism)	HLA class II	0.44	1.00

For the selection of peptides, lymph node metastasis from 2014 (tissue 3) was used. Variant allele frequency—proportion of reads which contain the variant allele (1 equals 100%). Patient’s HLA type—HLA-A*02:01, A*32:01, B*44:02, B*51:01, C*05:01, C*14:02.

## Data Availability

The data presented in this study are available on reasonable request from the corresponding author. The data are not publicly available because they contain information that could compromise the privacy of the research participants and third-party restrictions.

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
