# Peer review of "Case Report: Long-Term Survival of a Patient with Cerebral Metastasized Ovarian Carcinoma Treated with a Personalized Peptide Vaccine and Anti-PD-1 Therapy"

_vaccines, 2024, doi:10.3390/vaccines12040397_

Round 1

Reviewer 1 Report

Comments and Suggestions for Authors

The study is a case report focusing on immune responses of a patient with cerebral metastasized ovarian carcinoma treated with a personalized peptide vaccine and anti-PD-1 therapy. The study is interesting, well-performed and well-presented, however, some minor issues need to be addressed:

1. Did the authors detect T reg CD4+ cells in their immune monitoring/immunohistochemistry? it would be good to follow the T cell populations following peptide stimulation in Fig 2 and analyze more specific the immune responses.

2. The authors should discuss the limitations of their study in the Discussion section.

3. Did the patient benefit from the vaccination? Please discuss.

4. The finding about B2M loss is very interesting, the authors should discuss more the role of B2M loss in disease progression. There are several recent studies on this issue.

Author Response

Reviewer 1

The study is a case report focusing on immune responses of a patient with cerebral metastasized ovarian carcinoma treated with a personalized peptide vaccine and anti-PD-1 therapy. The study is interesting, well-performed and well-presented, however, some minor issues need to be addressed:

We thank Reviewer 1 for the positive feedback of our manuscript. Moreover, we appreciate the constructive suggestions that we have now addressed in the revised manuscript and below in the point-by-point response.

  1. Did the authors detect T reg CD4+ cells in their immune monitoring/immunohistochemistry? It would be good to follow the T cell populations following peptide stimulation in Fig 2 and analyze more specific the immune responses.

This would indeed be very interesting. Unfortunately, our assay cannot detect Regulatory T-cells. During the 12-day in-vitro expansion (in presence of cytokines IL-2 and IL-7), (neo)antigen-specific T-cells will significantly alter their surface marker and cytokine profiles. In order to detect Tregs, we would have to analyze them directly ex-vivo. Theoretically, this is possible for strong T-cell responses. However, in the presented case, we cannot do these experiments because we do not have PBMCs left.

For immunohistochemistry, it would be possible as well, but sadly again, we do not have material left from this patient. However, from other cancer patients undergoing this type of peptide vaccination, we observed that Treg-infiltration is very rare (only about 3% of all cells, compared to about 15% of non-regulatory T-Helper cells) and it is – so far – not associated with clinical or immunological responses.

  1. The authors should discuss the limitations of their study in the Discussion section.

We thank Reviewer 1 for raising this important point. We changed the following sentence, from:

“There are several emerging research questions that may warrant further studies in larger cohorts.”

To:

“There are several limitations that should be addressed in future studies. These include the ideal vaccine and adjuvants scheduling, …”

Limitations include the unstable vaccine schedule, usage of different adjuvants and the combination of different therapies. However, please note that this was an individual healing attempt after all other approved failed and the patient was faced with a fatal diagnosis. We do not think that it can be called a “study”.

  1. Did the patient benefit from the vaccination? Please discuss.

See also the comment above. The patient received different therapies at the same time, so the clinical success cannot definitely be attributed to our peptide vaccine. Still, what remains is the extraordinary long survival facing this life-threatening disease. We included the following sentence to the discussion:

“Although the patient received multiple therapies, and the clinical benefit cannot ultimately be attributed to neoantigen-derived peptide vaccination, strikingly, …”

  1. The finding about B2M loss is very interesting, the authors should discuss more the role of B2M loss in disease progression. There are several recent studies on this issue.

We thank Reviewer 1 for this positive feedback. We have minorly extended the discussion section regarding B2M loss by adding the following part into the discussion (minorly, because it is a single patient observation):

B2M gene mutations are enriched in uterine, breast, colorectal and gastric cancer and accumulate during cancer progression and targeted treatment (Wang et al. 2021). Accumulating evidence has shown that alterations of B2M gene and B2M proteins contribute to poor reaction to cancer immunotherapies by dampening antigen presentation (Tang et al. 2020). Hence, B2M is a potential biomarker for cancer immunotherapy (Reis et al. 2024).

Reviewer 2 Report

Comments and Suggestions for Authors The authors present an OC case with brain metastasis successfully treated with GM-CSF adjuvanted neoantigen peptide-based immunotherapy. This case is clinically encouraging because standard chemotherapy- highly resistant OC had been controlled by the therapeutic cancer vaccine for more than 4 years. The case presentation is clear and reliable.   My comments are as follows.   1. A characteristic feature of the case was robust response of a neoantigen-specific CD4 T cell. MKKS-pG52S is a 10 amino acid long CTL epitope peptide. What is the expected role of these MKKS-pG52S-specific CD4 T cells in antitumor immunity? CTL? Cytokine producing cells in the tumor microenvironment? Or Th to promote production of IgG antibodies against the antigen on the tumor cell surface?   2. During the period when the tumor was to be controlled (in 60-72 month), the case was treated with vaccine therapy as well as multiple other drugs including PD1Ab, Bev, Tamoxifen, and Carbo. The authors are recommended to discuss the roles of these drugs in the Discussion section. In addition, could the use of steroids do immunologically good for the vaccine therapy?   Minor points:   3 What is the histology of OC? 4. When the sample was obtained for B2M expression analysis?

Author Response

Reviewer 2

The authors present an OC case with brain metastasis successfully treated with GM-CSF adjuvanted neoantigen peptide-based immunotherapy. This case is clinically encouraging because standard chemotherapy- highly resistant OC had been controlled by the therapeutic cancer vaccine for more than 4 years. The case presentation is clear and reliable.   My comments are as follows. 

We thank Reviewer 2 for this encouraging statement. We have addressed the comments in the revised manuscript and in the point-by-point response below.  

  1. A characteristic feature of the case was robust response of a neoantigen-specific CD4 T cell. MKKS-pG52S is a 10 amino acid long CTL epitope peptide. What is the expected role of these MKKS-pG52S-specific CD4 T cells in antitumor immunity? CTL? Cytokine producing cells in the tumor microenvironment? Or Th to promote production of IgG antibodies against the antigen on the tumor cell surface?  

Although predicted to be a HLA class I binding peptide, MKKS-pG52S-specific T-cells clearly showed a CD4 phenotype. We observe that in approximately 9 % of all our vaccinated “short” peptides. The crucial role of CD8+ T-cells is of course well known, however, the ability of CD4+ T-cells to contribute to anti-tumor immunity independently of CD8+T cells is increasingly recognized (for example, nicely shown here: PMID: 37316667). We included this reference in the discussion section.

We “prefer” the hypothesis that these cells alter the tumor microenvironment and by cytokine release and favour and/or perform tumor cell killing. We do not think that antibodies play a major role (neoantigens are probably only intracellular; we might only induce antibodies against the vaccinated “linear peptide”; etc).  

  1. During the period when the tumor was to be controlled (in 60-72 month), the case was treated with vaccine therapy as well as multiple other drugs including PD1Ab, Bev, Tamoxifen, and Carbo. The authors are recommended to discuss the roles of these drugs in the Discussion section. In addition, could the use of steroids do immunologically good for the vaccine therapy?

See also the comment 3 from Reviewer 1. Yes, the patient received different therapies at the same time, so the clinical success cannot definitely be attributed to our peptide vaccine. Still, in this patient with ovarian cancer stage 4, metastasized to the brain, there is a very high likelihood of recurrence or progression in the first 2 years. As typical in the adjuvant or additive situation in a single case we do not know what exactly caused the good course in this individual patient. We have included the following sentence to the discussion section in order to dampen our statement:

“Although the patient received multiple therapies, and the clinical benefit cannot ultimately be attributed to neoantigen-derived peptide vaccination, strikingly, …”

  1. What is the histology of OC? 

We apologize for being not precise here. We included this information in the Case description:

We report the case of a female 49-year-old patient who was diagnosed with metastasized ovarian cancer in November 2010 (pT3b, nN1, M0; serous adenocarcinoma grading G3).

  1. When the sample was obtained for B2M expression analysis?

We analysed B2M expression by IHC in four samples and by NGS in two samples (see Figure 1 and “Case description”).

Reviewer 3 Report

Comments and Suggestions for Authors

In this article, the authors present a 49-year-old patient who was first diagnosed with stage III ovarian cancer. After the tumor progressed, the patient received a fully individualized neoantigen-derived peptide vaccine in the setting of an individual healing attempt. The manuscript is straightforward, well written, and concise and has clear results within the scope of a case report. Definitely deserves to be published and is a valuable contribution to the “vaccinesjournal. However, the following comments need to be addressed, as recommended.

[1] “Introduction”, Page 1 of 10, Lines 42-43:

OC is one of the most common cancers among women and the most lethal malignancy of all gynecological cancers (Reid et al. 2017; Sung et al. 2021).”.

Please, specify. Type I epithelial OCs are suggested to be relatively indolent and genetically stable tumors that typically arise from recognizable precursor lesions, such as endometriosis or borderline tumors with low malignant potential. In contrast, type II epithelial OCs are proposed to be biologically aggressive tumors from their outset, with a propensity for metastasis from small-volume primary lesions. High-grade serous – the most common type of epithelial OCs, accounting for approximately 75% of epithelial ovarian cancers – develop according to the type II pathway and present p53 and BRCA mutations.

Recommended reference: Pavlidis N, et al. The outcome of patients with serous papillary peritoneal cancer, fallopian tube cancer, and epithelial ovarian cancer by treatment eras: 27 years data from the SEER registry. Cancer Epidemiol. 2021;75:102045.

[2] “Introduction”, Page 2 of 10, Lines 55-57:

Hence, most patients with OC generally face a very poor prognosis, indicating that there is an urgent need for advanced therapeutic options.”.

Within this context, it is important to comment that PI3K pathway is frequently upregulated in epithelial OC and plays an important role in chemoresistance and preservation of genomic stability, as it is implicated in many processes of DNA replication and cell cycle regulation. The inhibition of the PI3K may lead to genomic instability and mitotic catastrophe through a decrease of the activity of the spindle assembly checkpoint protein Aurora kinase B and consequently increase of the occurrence of lagging chromosomes during prometaphase.

[3] “Introduction”, Page 2 of 10, Lines 60-63:

For OC, although promising results were obtained in preclinical and early phase clinical studies, the use of immunotherapeutic treatments has still not been implemented in clinical routine, mostly due to insufficient evidence of clinical effectiveness.”.

Nevertheless, the authors should mention the therapeutic strategy of the combinations of PARP inhibitors with immunotherapies, such as anti-CTLA-4 and PD-1/PD-L1 that has partly been based on the hypothesis that BRCA1/2, and wild-type BRCA1/2 homologous recombination (HR) deficiency tumours display a higher neo-antigen load than HR-proficient cancers, producing more effective anti-tumour immune response. In addition, there is evidence that BRCA deficiency may induce a STING-dependent innate immune response, by inducing type I interferon and pro-inflammatory cytokine production. Interestingly enough, clinical models have also demonstrated that PARP inhibition inactivate GSK3 and upregulate PD-L1 in a dose-dependent manner. Consequently, T-cell activation is being suppressed, resulting in enhanced cancer cell apoptosis.

[4] Discussion”, Page 8 of 10, Line 236:

However, most cancer patients still do not benefit from immunotherapeutic approaches.”.

This is not precise and should be rephrased. For instance, recently has been published that patients with melanoma of unknown primary (MUP) site seem to present better outcomes compared to those with stage-matched melanoma of known primary (MKP), probably due to higher immunogenicity as reflected in the immunologically mediated primary site regression. As such, MUP patients on immunotherapy display better outcomes when compared to the MKP site subset.

Comments on the Quality of English Language

Minor editing of English language required.

Author Response

Reviewer 3

In this article, the authors present a 49-year-old patient who was first diagnosed with stage III ovarian cancer. After the tumor progressed, the patient received a fully individualized neoantigen-derived peptide vaccine in the setting of an individual healing attempt. The manuscript is straightforward, well written, and concise and has clear results within the scope of a case report. Definitely deserves to be published and is a valuable contribution to the “vaccines” journal. However, the following comments need to be addressed, as recommended.

We thank Reviewer 3 for the positive feedback of our manuscript, and the evaluation of its relevance to the field. Moreover, we appreciate the constructive criticism that we have now addressed in the revised manuscript and below in the point-by-point response.

[1] “Introduction”, Page 1 of 10, Lines 42-43:

“OC is one of the most common cancers among women and the most lethal malignancy of all gynecological cancers (Reid et al. 2017; Sung et al. 2021).”.

Please, specify. Type I epithelial OCs are suggested to be relatively indolent and genetically stable tumors that typically arise from recognizable precursor lesions, such as endometriosis or borderline tumors with low malignant potential. In contrast, type II epithelial OCs are proposed to be biologically aggressive tumors from their outset, with a propensity for metastasis from small-volume primary lesions. High-grade serous – the most common type of epithelial OCs, accounting for approximately 75% of epithelial ovarian cancers – develop according to the type II pathway and present p53 and BRCA mutations.

Recommended reference: Pavlidis N, et al. The outcome of patients with serous papillary peritoneal cancer, fallopian tube cancer, and epithelial ovarian cancer by treatment eras: 27 years data from the SEER registry. Cancer Epidemiol. 2021;75:102045.

We apologize for being unprecise here. We have extended the introduction according to the Reviewers suggestion and included the recommended reference.

[2] “Introduction”, Page 2 of 10, Lines 55-57:

“Hence, most patients with OC generally face a very poor prognosis, indicating that there is an urgent need for advanced therapeutic options.”.

Within this context, it is important to comment that PI3K pathway is frequently upregulated in epithelial OC and plays an important role in chemoresistance and preservation of genomic stability, as it is implicated in many processes of DNA replication and cell cycle regulation. The inhibition of the PI3K may lead to genomic instability and mitotic catastrophe through a decrease of the activity of the spindle assembly checkpoint protein Aurora kinase B and consequently increase of the occurrence of lagging chromosomes during prometaphase.

We thank the reviewer for this comment and apologize for being unclear here. Our intention in this part was to emphasize that prognosis for advanced stages is poor (regardless which pathways are up- or downregulated) and that unfortunately most OC are firstly diagnosed already in advanced stages. We changed this phrase to:

“As a result, most patients diagnosed with OC face a relatively poor prognosis, indicating that there is an urgent need for advanced therapeutic options.”

[3] “Introduction”, Page 2 of 10, Lines 60-63:

“For OC, although promising results were obtained in preclinical and early phase clinical studies, the use of immunotherapeutic treatments has still not been implemented in clinical routine, mostly due to insufficient evidence of clinical effectiveness.”.

Nevertheless, the authors should mention the therapeutic strategy of the combinations of PARP inhibitors with immunotherapies, such as anti-CTLA-4 and PD-1/PD-L1 that has partly been based on the hypothesis that BRCA1/2, and wild-type BRCA1/2 homologous recombination (HR) deficiency tumours display a higher neo-antigen load than HR-proficient cancers, producing more effective anti-tumour immune response. In addition, there is evidence that BRCA deficiency may induce a STING-dependent innate immune response, by inducing type I interferon and pro-inflammatory cytokine production. Interestingly enough, clinical models have also demonstrated that PARP inhibition inactivate GSK3 and upregulate PD-L1 in a dose-dependent manner. Consequently, T-cell activation is being suppressed, resulting in enhanced cancer cell apoptosis.

We included information which approaches are specifically tested. However, we would like to refrain from going more into detail for example regarding BRCA deficiencies, etc because our focus was on checkpoint inhibition and cancer vaccination and why personalized approaches might work better.

[4] “Discussion”, Page 8 of 10, Line 236:

“However, most cancer patients still do not benefit from immunotherapeutic approaches.”.

This is not precise and should be rephrased. For instance, recently has been published that patients with melanoma of unknown primary (MUP) site seem to present better outcomes compared to those with stage-matched melanoma of known primary (MKP), probably due to higher immunogenicity as reflected in the immunologically mediated primary site regression. As such, MUP patients on immunotherapy display better outcomes when compared to the MKP site subset.

We apologize for being not precise here. We rephrased the sentence as follows:

“However, many studies in recent years showed that only subgroups of patients are benefitting from immunotherapeutic approaches. Among many other strategies, personalized vaccination…”